# Unveiling the Impact of Gene Presence/Absence Variation in Driving Inter-Individual Sequence Diversity within the CRP-I Gene Family in *Mytilus* spp.

**DOI:** 10.3390/genes14040787

**Published:** 2023-03-24

**Authors:** Nicolò Gualandi, Davide Fracarossi, Damiano Riommi, Marco Sollitto, Samuele Greco, Mario Mardirossian, Sabrina Pacor, Tiago Hori, Alberto Pallavicini, Marco Gerdol

**Affiliations:** 1Area of Neuroscience, International School for Advanced Studies, 34136 Trieste, Italy; ngualand@sissa.it; 2Department of Life Sciences, University of Trieste, 34127 Trieste, Italy; davide.fracarossi@studenti.units.it (D.F.); damiano.riommi@studenti.units.it (D.R.); marco.sollitto@famnit.upr.si (M.S.); samuele.greco@units.it (S.G.); mmardirossian@units.it (M.M.); pacorsab@units.it (S.P.); pallavic@units.it (A.P.); 3Faculty of Mathematics, Natural Sciences and Information Technologies, University of Primorska, 6000 Koper, Slovenia; 4Atlantic Aqua Farms Ltd., Vernon Bridge, PE C0A 2E0, Canada; tiago.hori@atlanticaquafarms.com; 5Anton Dohrn Zoological Station, 80121 Naples, Italy

**Keywords:** defense peptides, innate immunity, gene presence/absence variation, cysteine-rich

## Abstract

Mussels (*Mytilus* spp.) tolerate infections much better than other species living in the same marine coastal environment thanks to a highly efficient innate immune system, which exploits a remarkable diversification of effector molecules involved in mucosal and humoral responses. Among these, antimicrobial peptides (AMPs) are subjected to massive gene presence/absence variation (PAV), endowing each individual with a potentially unique repertoire of defense molecules. The unavailability of a chromosome-scale assembly has so far prevented a comprehensive evaluation of the genomic arrangement of AMP-encoding loci, preventing an accurate ascertainment of the orthology/paralogy relationships among sequence variants. Here, we characterized the CRP-I gene cluster in the blue mussel *Mytilus edulis*, which includes about 50 paralogous genes and pseudogenes, mostly packed in a small genomic region within chromosome 5. We further reported the occurrence of widespread PAV within this family in the *Mytilus* species complex and provided evidence that CRP-I peptides likely adopt a knottin fold. We functionally characterized the synthetic peptide sCRP-I H1, assessing the presence of biological activities consistent with other knottins, revealing that mussel CRP-I peptides are unlikely to act as antimicrobial agents or protease inhibitors, even though they may be used as defense molecules against infections from eukaryotic parasites.

## 1. Introduction

The term “mussel” is generally used to indicate a relatively large and heterogeneous group of over 80 bivalve mollusk genera, mostly living in marine environments. Not to be confused with the so-called “freshwater mussels”, distantly related bivalves belonging to the order Unionida (infraclass Heteroconchia), true mussels are taxonomically classified in the order Mytilida (infraclass Pteriomorphia) [1]. *Mytilus* Linnaeus, 1758 is one of the most common mussel genera, which displays cosmopolitan distribution, with locally highly abundant populations in temperate marine coastal areas. Although the exact taxonomic boundaries among different *Mytilus* species are still widely debated in the scientific community due to incomplete reproductive isolation, this genus currently includes several interfertile species. Excluding the divergent *Mytilus californianus* Conrad, 1837 and *Mytilus unguiculatus* Valenciennes, 1858, the *Mytilus* species complex includes *Mytilus edulis* Linnaeus, 1758, *Mytilus galloprovincialis* Lamarck, 1819 and *Mytilus trossulus* Gould, 1850, native to the Northern hemisphere, plus *Mytilus chilensis* Hupé, 1854, *Mytilus platensis* d’Orbigny, 1842, *Mytilus planulatus* Lamarck, 1819 and *Mytilus aoteanus* Powell, 1958 [2,3], native to the Southern hemisphere. Mussels have been part of the human diet in coastal populations for several centuries. While traditional harvesting methods continue today, mussels are also considered among the most relevant aquacultured bivalves, along with oysters, clams and scallops, as their global estimated production exceeds 250 K tons/year [4].

Besides their commercial importance, mussels are frequently used as sentinel organisms in biomonitoring programs due to their high tolerance to biotic and abiotic stress, as well as their bioaccumulation ability [5]. However, the latter feature also represents a source of concern for edible mussels’ production due to the possible impact of accumulated biotoxins, heavy metals and other pollutants on human health [6,7]. Mussels also show a remarkable resistance towards pathogenic infections causing massive mortality events in other bivalve species [8,9], displaying increased tolerance following repeated exposures to the same pathogen [10].

Although the reasons behind this remarkable ability to withstand environmental alterations have not been fully clarified yet, the development of a highly diversified and complex immune system has been assumed to be a key factor of their resistance ever since the earliest molecular investigations [11]. Indeed, mussels produce a particularly large and intricate repertoire of immune molecules, ranging from soluble, membrane-bound and intracellular Pattern Recognition Receptors (PRRs), to signaling mediators and effectors with antiviral, antibacterial, antifungal or antiprotozoal action [12]. Our knowledge of this large arsenal of immune-related gene products has continued growing at a very fast rate during the past couple of decades. The recent release of the complete genome sequence of the Mediterranean mussel *M. galloprovincialis* provided a major breakthrough towards the understanding of this molecular diversity [13]. Indeed, the comparative analysis of different individuals revealed that a significant fraction of mussel protein-coding genes were not shared by all individuals, i.e., they were dispensable and subjected to presence/absence variation (PAV). This phenomenon, previously well-documented only in prokaryotes, plants and fungi, but nearly unexplored in metazoans, mostly targeted largely expanded gene families involved in immune response and survival, with possible implications in explaining the great capability of environmental adaptation and invasiveness of this species.

Strikingly, the dispensable fraction of the mussel genome was strongly enriched in genes encoding short secretory defense peptides. Such molecules have been the subject of intense scrutiny since the early ‘90s due to their high expression in hemocytes, circulating cells with marked phagocytic activity, thought to act as the main players in the context of bivalve immune response [14,15]. The forerunner studies carried out on this topic revealed that hemocyte-associated mussel defense peptides are cysteine-rich, bear a positive net charge and belong to four distinct families, i.e., defensins, mytilins, myticins and mytimycins. Although these peptides displayed some degree of antimicrobial or antifungal activity, thereby granting a classification as antimicrobial peptides (AMPs) [16], recent studies challenged this univocal interpretation, suggesting that some isoforms may act as immunomodulatory molecules in a cytokine-like fashion [17]. Following these early studies, the increasing recognition of the importance of mucosal surfaces in bivalve immune response [18] led to the discovery of several other families of cysteine-rich defense peptides, as well as a few linear AMPs [19], displaying markedly different tissue specificity, i.e., expressed in the mantle, gills or digestive gland tissues. These include big defensins, mytimacins [20], myticusins [21], pseudomytilins [22] and the most recent addition to the list, three novel distinct families of defensin-like peptides showing a cysteine-stabilized αβ motif [23]. However, the repertoire of mussel short secretory cysteine-rich peptides might be even larger, as suggested by the large fraction of taxonomically restricted gene families found in the *Mytilus* genus [13] and by the previous identification, by our group, of CRP-I (Cysteine-Rich Peptide family I), a functionally uncharacterized novel family of hypervariable cationic peptides with six highly conserved cysteine residues [24].

Literature data have always been consistent in identifying a remarkable primary sequence diversity and inter-individual variation of expression as distinctive features of mussel AMPs. Nevertheless, the interpretation of these observations varied quite significantly from study to study, proposing the presence of responder and non-responder individuals, the influence of hidden environmental factors, extreme allele diversity, gene conversion or somatic mutation as alternative, non-mutually exclusive explanations [25,26,27]. The recent release of high quality genomic data from multiple *Mytilus* species, as well as from different individuals belonging to the same species, has finally clarified that gene PAV is the main driver of these otherwise unexplained patterns [28,29,30]. Nevertheless, a complete overview of the architecture of the loci encoding these defense peptides is still lacking.

In the Mediterranean mussel, the members of the CRP-I family are characterized by a remarkable sequence diversity in the mature peptide region, in stark contrast with the high conservation of the signal peptide and propeptide regions [24]. Although we could not conclusively establish the biological function of these peptides in our previous work, we showed that CRP-I displayed a C(X_3–6_)C(X_1–7_)CC(X_3–4_)C(x_3–5_)C cysteine array, suggesting a knottin-like fold structural arrangement. The knottin superfamily includes highly stable peptides, which share a similar three-dimensional structure, stabilized by three highly conserved disulfide bonds, found in all domains of life, as their typical folding was independently acquired by convergent evolution in multiple unrelated phyla [31]. The knottin folding is often associated with bioactive proteins, acting as AMPs [32], toxins [33] or protease inhibitors [34]. In our first work on this subject, the characterization of mussel CRP-I peptides was undoubtedly hampered by the lack of complete genomic resources, by the fact that the impact of gene PAV on inter-individual sequence diversity was still unknown, and by the unavailability of reliable ab initio structural prediction methods [24].

In this study, we aimed to fill these knowledge gaps, investigating the genomic architecture of the CRP-I gene locus in the fully-phased chromosome-scale genome assembly of *M. edulis* and assessing the impact of gene PAV in the *Mytilus* species complex. Moreover, based on modern deep learning-based structural prediction methods, we provide further evidence that CRP-I peptides adopt a knottin-like fold, which prompted us to evaluate the possible biological function of the representative peptide sCRP-I H1.

## 2. Materials and Methods

### 2.1. Characterization of the M. edulis CRP-I Genomic Loci

The chromosome-scale genome assembly of the blue mussel *M. edulis* (version PEIMed) was analyzed with the aim to identify and manually annotate all CRP-I encoding genes. Taking into account the previously reported pan-genomic structure of *Mytilus* genomes [13], we proceeded with an in-depth analysis of an alternative fully-phased version of the blue mussel genome assembly, improved by exploiting the information deriving from chromosome conformation capture libraries, which allowed the investigation of the two separately assembled haplotypes. The technical details of the chromosome phasing process are provided in Appendix A.

The annotation of CRP-I genes was performed as follows. In detail, all the full-length precursor sequences of previously reported CRP-I proteins [24] were used as queries for tBLASTn searches, which were initially based on an arbitrarily set e-value threshold equal to 0.05 against the PEIMed reference. Based on the known exon/intron organization of CRP-I genes, this allowed us to preliminarily identify the approximate coordinates of the protein coding regions within exons 2, 3 and 4. These initial hits were further refined as follows. First, matching mRNA sequences from available transcriptomic datasets from *Mytilus* spp. retrieved from the NCBI TSA datasets, or reported in our previous study [24], were aligned against the corresponding genomic sequence (extended by 5 Kb at both the 5’ and 3’ ends) with MUSCLE [35], allowing the precise identification of putative exons and introns. This process allowed the inclusion of 5’ (i.e., the complete exon 1 and part of exon 2) and 3’ UTRs (part of exon 4), whenever possible. If no perfect match was available, the most similar available sequence was used as a replacement. Splicing acceptor and donor sites were subsequently refined with Genie [36] and the resulting predicted mRNAs were virtually translated to proteins with the aim to identify the coding sequence and to verify the absence of in-frame stop codons or frameshift mutations.

CRP-I pseudogenes were called based on the detection of at least one of the five following warning flags: (i) At least one of the three exons spanning the ORF was missing, the lack of the short exon 1, only including 5’ UTR sequence, was ignored due to the inherent difficulty of its detection; (ii) The predicted precursor protein displayed in-frame stop codons or frameshift mutations; (iii) The encoded mature peptide lacked one or more cysteine residues of the knottin array; exceptions were made for gene models supported by transcriptomic evidence; (iv) The predicted precursor protein lacked a detectable dibasic propeptide cleavage site (i.e., KR or RR); (v) The predicted gene model showed no detectable canonical splicing donor or acceptor sites.

The genes resulting from this manual annotation process that displayed a complete ORF with no warning flags were classified as s- (single cysteine array) or m- (multiple cysteine array) CRP-I depending on whether a single or multiple C-C-CC-C-C motifs were identified within exon 4. The manual annotation process was then re-iterated by adding newly annotated CRP-I genes, until no further hits could be detected.

The same annotation strategy was used on the phased blue mussel genome assembly, identifying the CRP-I genes and pseudogenes associated with either of the two haplotypes. The corresponding scaffolds were subsequently attached to matching coordinates in the monoploid reference assembly based on nucleotide sequence homology.

### 2.2. Collection of CRP-I Sequences in the Mytilus Species Complex

All available genome assemblies for the species belonging to the *Mytilus* species complex were screened using the strategy described in Section 2.1. The target genomes included the Mediterranean mussel reference (hereafter named LOLA), the 14 resequenced genomes reported in the same paper (GALF1, GALF2, GALF3, GALM1, GALM2, GALM3, GALM6, GALM11, ITAF1, ITAF2, ITAF3, ITAM1, ITAM2 and ITAM3) [13] and the draft genome assemblies reported by Murgarella et al. (PURA) [37], Nguyen et al. (MgalAUS) [38] and Simon (MgalMED) [39]; the three available alternative blue mussel genome assemblies from Corrochano-Fraile et al. and Simon (MeduMEDL1, MeduEUN and MeduEUS) [39,40]; the reference genome assembly of *M. chilensis* [41].

Due to the variable quality of these assemblies, only gene models including a complete ORF and with no warning flags were subjected to further consideration. Whenever no full gene model could be recovered, the corresponding full-length CDS was extracted from de novo assembled transcriptomes (obtained from NCBI SRA data). Partial gene models unsupported by RNA-seq data were discarded.

To simplify the addition of novel sequences to this family in the future, the nomenclature of the full-length sequences was established as follows, replacing the one defined in our previous publication [24]. First, based on the observation that several identical amino acid sequences were present in different species of the *Mytilus* complex, consistently with known genetic introgression patterns [42], the use of the prefix Mg (to indicate *M. galloprovincialis*) was deprecated. Moreover, we opted to uniquely base nomenclature on the amino acid sequence of the mature peptide, disregarding minor non-synonymous changes occurring in the signal peptide and propeptide regions, unlikely to affect the biological function of the active mature CRP-I peptides. Entries from our previous publication that referred to pseudogenes or unconfirmed partial genes were removed, as summarized in Appendix A. All confirmed CRP-I sequences were subsequently assigned a unique identifying code, composed of either an uppercase letter (for sCRP-I peptides) or by a Greek alphabet letter (for mCRP-I peptides) to design hypothetical groups of allelic variants and/or nearly identical paralogous gene copies, followed by a number, to design the different variants belonging to such groups. These assignments were strictly based on pairwise homology and phylogenetic criteria, as explained in Section 2.3.

### 2.3. CRP-I Evolution and Sequence Analysis

The open reading frames of each s- and mCRP-I sequence were translated to the encoded protein sequences, which were subsequently analyzed to identify the signal peptide and proprotein convertase cleavage site with SignalP v.6.0 and ProP v.1.0, respectively [43,44]. The isoelectric point and molecular weight of the predicted mature peptides were calculated with IPC v.2.0 [45] and the spacing between the six highly conserved cysteine residues was used to define ten distinct cysteine array types (indicated with Roman numerals, i.e., I–X).

All sCRP-I and mCRP-I full precursor sequences were separately aligned with MUSCLE [36]. Based on preliminary tests, the sequences lacking four (i.e., group E) or all (group λ) cysteine residues were added to the sCRP-I and mCRP-I MSAs, respectively. The resulting multiple sequence alignments (MSAs) were manually refined, whenever needed, to adjust the position of the cleavage sites and cysteine residues, as well as to match exon–exon junctions, based on available information on gene organization. The signal peptide and propeptide regions were removed, and the MSA of mCRP-I sequences was truncated to only include the first cysteine array. Then, the Hamming dissimilarity matrix for all the mature sCRP-I sequences was computed (including gaps) and used to define 23 groups (A-W) based on a pairwise distance threshold of seven. The mCRP-I sequences were classified with a slightly different approach, performing pairwise comparisons on each individual cysteine array, defining 16 groups (α-π).

The MSAs of the full precursor sequences were used as inputs for maximum likelihood (ML) phylogenetic inference analyses with W-IQ-TREE [46]. In detail, the best-fitting models of molecular evolution were identified, with ModelFinder [47], according to the Bayesian Information Criterion, as JTTDCMut + I + G4 and FLU + I + G4 for the sCRP-I and mCRP-I datasets, respectively. The reliability of the trees was tested with 1000 ultrafast bootstrap replicates. The consistency of the aforementioned sequence groups was evaluated by verifying that all their members were part of monophyletic clades supported by bootstrap values >80. Exceptions were arbitrarily made for sequences characterized by the presence of very long branches or, in the case of mCRP-I, displaying a different number of cysteine arrays.

### 2.4. Investigation of Gene PAV Patterns

Gene PAV was investigated in all available genomes for the species belonging to the *Mytilus* complex (i.e., 18 *M. galloprovincialis*, 4 *M. edulis* and 1 *M. chilensis* individuals). To take into account the impossibility of discriminating paralogous genes from allelic variants in the absence of high quality genome assemblies for most of these species. PAV patterns were studied for each of the aforementioned groups of hypothetical allelic variants and/or recently duplicated paralogous genes (i.e., A-W for sCRP-I and α-π for mCRP-I). Gene PAV data were computed based on the presence of potentially functional gene sequences, which means that truncated or otherwise likely non-functional pseudogenes were treated separately. We calculated the dispensability index (DI) for each sequence group, defined as the fraction of resequenced *Mytilus* genomes where they were absent.

### 2.5. Evaluation of Tissue Specificity

The extreme levels of primary sequence diversity and widespread gene PAV observed in the CRP-I family do not presently allow an accurate estimate of gene expression levels by qRT-PCR. Similarly, the in-silico calculation of expression levels based on the mapping of high throughput RNA-seq datasets against a reference genome would be hampered by the highly diverse repertoire of CRP-I genes carried by each individual. Based on these assumptions, we developed a bioinformatics approach to allow a reliable quantification of gene expression. This analysis, carried out for each sequence group, used 215 publicly available high-quality transcriptomic datasets from four tissues of *M. galloprovincialis*, i.e., digestive gland, mantle, gills and hemocytes, and mid-trochophore stage larvae (Appendix A).

Briefly, the full nucleotide sequence of the ORF of all CRP-I sequences were used as a reference for read mapping. A set of ten housekeeping genes (accession IDs: *MGAL10A075356*—60S ribosomal protein L32, *MGAL10A009412*—60S acidic ribosomal protein P2, *MGAL10A086281*—60S ribosomal protein L14, *MGAL10A086168*—60S ribosomal protein L34, *MGAL10A044516*—40S ribosomal protein S19, *MGAL10A041348*—60S ribosomal protein L11, *MGAL10A061238*—60S ribosomal protein L18a, MGAL10A087486—60S ribosomal protein L7a, *MGAL10A049522*—40S ribosomal protein S21, *MGAL10A090487*—40S ribosomal protein S5a), selected based on their high stability of expression in RNA-seq experiments carried out in the Mediterranean mussel [48], was used as a reference for calculations. In detail, following quality trimming, RNA-seq data was mapped with the CLC Genomics Workbench v.22 against the sequence list comprising the reference genes and all CRP-I sequences. Mapping parameters were set to 0.75 (length fraction) and 1 (similarity fraction), respectively. Read counts obtained from each CRP-I sequence, normalized by the length of the sequence, were cumulatively reported for each of the sequence groups defined in Section 2.3. Such values were converted to TPM, using the average expression values obtained from the reference genes, assumed to represent a highly stable metric for all tissues and experimental conditions (i.e., roughly 3000 TPM [13]) as a normalizing factor.

### 2.6. Structural Prediction

Alphafold v2.3.0 [49,50] was used to predict the 3D structures of the mature peptides of all sCRP-I and mCRP-I proteins, with CASP14-like settings (--model_preset = monomer_casp14 and --db_preset = full_dbs --max_template_date = 2022-11-15), granting the highest level of accuracy and reproducibility. Each predicted model was accompanied by per-residue confidence estimates, reported on a scale from 0 to 100 based on pLDDT [51]. The presence of disulfide bonds was detected with ChimeraX [52], using the “bond” function for every cysteine residue (select::name = “CYS”; bond sel). Dali [53] was used to identify the peptides displaying the most similar 3D structures within the PDB25 database, considering the results with z-score >2 as indicative of non-spurious structural overlap.

### 2.7. Functional Characterization of the sCRP-I H1 Peptide

The mature peptide sCRP-I H1 (i.e., APCWPRGCFRDRDCCYGYQCSYRKCMRKR-NH_2_, assuming the occurrence of C-terminal amidation, a common post-translational modification performed by α-amidating monooxygenase in molluscan short peptides [54]) was selected for solid phase synthesis as a representative member of the family. Peptide synthesis was carried out by Synpeptide (Shanghai, China) using F-moc solid phase synthesis, and subsequently directing the formation of disulfide bonds in the typical connectivity found in knottins (i.e., Cys_1_-Cys_4_, Cys_2_-Cys_5_, Cys_3_-Cys_6_). The synthetic peptide was subjected to multiple functional assays, aimed at evaluating its likelihood of carrying out biological activities similar to other functionally characterized knottins.

#### 2.7.1. Protease Inhibitor Activity

Protease stock solutions were prepared at 1 mg/mL in H_2_O for papain, thermolysin and subtilisin A, and at 0.1 mM HCl for pepsin. Stock solutions were diluted in the appropriate buffer solution (see below) at a final concentration of 1 μg/mL for papain, thermolysin and subtilisin A. Pepsin was diluted to a final concentration of 10 μg/mL. Buffers were composed as follows: 30 mM CaCl_2_, 0.1% (*w*/*v*) Brij e 20% (*w*/*v*) sucrose, pH 7.9 for thermolysin and subtilisin A; 150 mM Tris–HCl, 300 mM NaCl, 0.1% (*w*/*v*), Brij, 2.0 mM EDTA, 5.0 mM cysteine and 20% (*w*/*v*) sucrose, pH 6.2 for papain. Tris–HCl at 150 mM, 30 mM CaCl_2_, 0.1% (*w*/*v*) Brij, 20% (*w*/*v*) sucrose, pH 2.0 for pepsin. Protease working solutions were then prepared at a final concentration of 1.37 μM in each specific buffer. For each protease, 20 μL were then incubated for 30 min at room temperature (~25 °C) with 10 μL of sCRP-I H1 solution or with 10 μL of buffer (used as control). Solutions were tested in 96 well plates adding, as protease substrate, 2.5 mg of Hide-Powder Remazol Brilliant Blue R (Sigma-Aldrich, Saint Louis, MO, USA) with 100 μL of buffer solution. Incubation of 96 well plates was at 37 °C, with shaking, for 45 min for papain, subtilisin A and thermolysin, and for 1 h for pepsin. These incubation times were defined based on preliminary tests carried out without the addition of the target peptide. Plates were centrifuged for 5 min at 800 g, allowing the sedimentation of the unprocessed substrate. Supernatant absorbance was then read at 595 nm (Tecan, Mannedorf, Zurich, Switzerland). Experiments were carried out in triplicates.

#### 2.7.2. Antimicrobial Activity

The potential of sCRP-I H1 for inhibiting bacterial growth was assessed on four different bacterial strains: *Escherichia coli* ATCC 25922, *Staphylococcus aureus* ATCC 25923, *Pseudomonas aeruginosa* ATCC 27853 and *Enterococcus faecalis* ATCC 19433. All bacterial strains were cultured at 37 °C in Mueller–Hinton (MH) medium. MH was prepared, according to the manufacturer’s instructions, dissolving 21 g of Difco MH medium powder (Thermo Fisher Scientific, Waltham, MA, USA) in 1L H_2_O. Agar was added at 15 g/L in medium to prepare solid MH medium. The medium was sterilized by autoclaving. Bacterial strains were conserved in glycerol stocks at −80 °C and cultivated on solid medium overnight at 37 °C to allow colony formation. Bacterial suspensions were then prepared from three or four colonies by inoculating them in 5 mL of fresh liquid MH medium. Bacterial suspensions were then grown overnight at 37 °C with shaking (130 rpm). The following day, 200 μL of overnight bacterial cultures were added to 10 mL of new MH broth and grown at 37 °C, with shaking (130 rpm), for approximately 2 h, up to their mid logarithmic growth phase (approximately OD_600nm_ = 0.3). The bacterial growth phase was assessed using the following conventions: OD_600nm_ = 0.31 corresponding to 4.6 × 10^7^ CFU/mL for *E. coli*; OD_600nm_ = 0.1 corresponding to 10^8^ CFU/mL for *S. aureus*; OD_600nm_ = 0.3 corresponding to 10^7^ CFU/mL for *P. aeruginosa*; OD_600nm_ = 0.3 corresponding to 3.7 × 10^8^ CFU/mL for *E. faecalis*. For MIC calculation, the sCRP-I H1 peptide was dissolved in MH and serially two-fold diluted in the wells of microtiter round-bottom 96 well plates (Sarstedt, Nümbrecht, Germany) in a final volume of 50 μL, testing the concentrations of 32 μM, 16 μM, 8 μM, 4 μM, 2 μM, 1 μM, 0,5 μM. A volume of 50 μL of bacterial suspension (5 × 10^5^ UFC/mL) was then added to each well to obtain a final concentration of 2.5 × 10^5^ CFU/mL. Plates were then sealed with parafilm, to reduce evaporation, and incubated at 37 °C overnight (the total incubation time was approximately 24 h). MIC was calculated as the lowest concentration of peptide resulting in the complete inhibition of visible bacterial growth in the wells after overnight incubation. All tests were carried out in independent triplicates.

#### 2.7.3. In Vivo Toxicity Test

The peptide was dissolved to the desired concentration in a final volume of 50 μL of sterile Phosphate Buffered Saline (PBS) and injected, using a sterile syringe, in *Galleria mellonella* larvae (average weight ~0.5 g). The control group, composed of 65 individuals, was injected with 50 μL of sterile PBS, pre-filtered with 0.22 μm filters, while the killing positive control group was injected with 50 μL glutaraldehyde. The two test groups, each composed of 65 individuals, were injected, respectively, with 300 e 30 μg di peptide/Kg (800 e 80 pmol/g, respectively) for a total injection volume of 50 μL. Larvae were monitored for neurotoxicity signs for 48 h, including lack of movement, twitching, death and melanization. During the experimental time course, larvae were not fed and kept at room temperature.

#### 2.7.4. MTT Assay

To determine the possible cytotoxic effect of the synthetic peptide, the colorimetric 3-(4,5-dimethylthiazol-2-yl)-2,5-diphenyl tetrazolium bromide (MTT) assay was performed. The SH-SY5Y cell line was cultured in Dulbecco’s modified Eagle medium (DMEM) (Euroclone, Pero, Italy) supplemented with 10% (*v*/*v*) fetal bovine serum (FBS) (Euroclone, Pero, Italy). Cells were seeded in the wells of a flat-bottom 96 well plates at a density of 2 × 10^4^ cells per well in a volume of 90 μL and incubated for 24 h at 37 °C in 5% CO_2_. Different sCRP-I H1 peptide concentrations, dissolved in DMEM, were added to each well to reach a final concentration of 100 μM, 50 μM, 25 μM, 10 μM, 1 μM, 0.5 μM, respectively. After 20 h of treatment, 20 μL of a MTT solution (5 mg/mL in PBS) was added to each well. After 4 h of incubation in the dark at 37 °C at 5% CO_2_, 100 μL of Igepal (10% *w*/*v* in 10 mM HCl) was added to each well. The plate was incubated overnight at 37 °C at 5% CO_2_ to allow the solubilization of the crystals. Absorbance was then assessed at 544 nm using a microplate reader.

## 3. Results and Discussion

### 3.1. An Updated Catalogue of Mussel CRP-I Precursors

The new strategy of retrieval of CRP-I sequences from *Mytilus* spp. genomes led to a significant update of the repertoire of these cysteine-rich peptides compared with our previous publication, where many out of the 51 sCRP-I and 16 mCRP-I reported sequences were incomplete [24]. The strategy outlined in Section 2.1 clarified that several of these were pseudogenes (see further discussion in Section 3.4 below), which were deprecated in this paper, together with other variants linked with minor polymorphism that did not affect the mature peptide region. As detailed in Appendix A, this process led to the removal of 15 sCRP-I and 7 mCRP-I sequences, respectively, and to the update of another eight partial or mis-annotated sequences (four sCRP-I and four mCRP-I) to their full-length precursors. In summary, the final sequence collection included 114 sCRP-I and 82 mCRP-I sequences (Appendix A). These included four precursor peptides lacking all the six expected cysteine residues and a single sequence only retaining the Cys_3_/Cys_4_ pair. These were classified, based on phylogenetic and genomic criteria, as degenerated sequences belonging to the sCRP-I (see Section 3.1.1) and mCRP-I (see Section 3.1.2) subfamilies, respectively.

#### 3.1.1. Molecular Diversity and Evolution of the sCRP-I Subfamily

The clustering criteria detailed in Section 2.3 allowed the categorization of sCRP-I sequence within 23 distinct sequence groups (A-W) sharing high pairwise primary sequence homology. With the lone exception of the single peptide belonging to group E, these sequences shared all previously described expected features of functional members of the CRP-I family, i.e., the presence of a signal peptide, a well-conserved dibasic (i.e., KR or RR) proprotein convertase cleavage site and six invariant cysteine residues, defining a C-C-CC-C-C array. sCRP-I groups included a highly variable number of sequences, ranging from one (groups E, Q, R, T and U) to 11 (groups H and P), highlighting different evolutionary dynamics. In the absence of genomic data, this remarkable intra-group sequence diversity could be either interpreted as the product of allelic variation or, alternatively, as the presence of several similar paralogous genes. This aspect will be further discussed in Section 3.3.

As previously reported [24], sCRP-I precursor proteins are characterized by an extreme primary sequence diversity within the mature, cysteine-rich peptide region, which is subjected to positive selection, and by highly conserved signal peptide and propeptide regions. This is exemplified by the MSA between representative members of the 23 sequence groups (Figure 1A; the full MSAs for each group are reported in Appendix A). Although sCRP-I peptides shared a highly conserved cysteine array, the spacing between the six cysteine residues was highly variable, as evidenced by the identification of ten distinct types of arrays (indicated by roman numerals, i.e., I-X, Figure 1B). The length of the loops connecting the conserved cysteine residues in the knottin superfamily, i.e., the structural fold CRP-I peptides belong to (see Section 3.5 for further discussion), is subjected to structural constraints [31]. We assessed the consistency of whether the size of such loops in sCRP-I peptides was consistent with the range observed in other members of the knottin superfamily. The lengths of loops (3–6 aa), loop d (4 aa in most sequence groups) and loop e (3–5 aa) were well within expectations. In 90% of known knottins, the length of loop b is comprised of between two and seven aa. This loop was extremely short (just a single aa) in the sCRP-I sequences belonging to group B, and included either four or five residues in all other sCRP-I groups, except group O (7 aa). Hence, only group B sequences did not match expectations (the relevance of this observation with respect to the structure of the knottin fold will be further discussed in Section 3.5).

This updated analysis of sCRP-I mature peptides confirmed their previously reported general cationic charge [24], averaging an isoelectric point of 8.43. Nevertheless, significant variation was observed among groups, with pI ranging from over ten (groups E, G and P) to less than four (group Q) (Figure 1C). The molecular weight was also within previously reported ranges, averaging 3.56 kDa; with the exception of the degenerated group E peptide, the smallest sCRP-I peptides belonged to group B (i.e., ~2.62 kDa), in line with their short loop b. On the other hand, the largest sCRP-I peptide was R1, with 4.33 kDa (Figure 1D). Such estimates do not take into account the possibility that mature peptides may undergo post-translational modifications, such as C-terminal amidation, whose occurrence should be experimentally verified in the future.

The monophyly of the 23 sCRP-I groups was highly supported (bootstrap values > 80) in all cases, with the lone exception of group V, which appeared to be “nested” within group S (Figure 2A). Strikingly, the classification of the sCRP-I sequences identified in this study was also well-supported by the primary sequence distance among mature peptides, which led to the placement of each group within a well-defined space in the MDS plot (Figure 2B). Although the bootstrap support of some basal nodes of the phylogenetic tree was not particularly high, which may be explained by the short length and high diversity of the analyzed sequences, the topology of the phylogenetic tree clearly identified the evolutionary relationships among different sequence groups. Phylogeny also recapitulated the presence of the same disulfide array type (Figure 1B) in closely related CRP-I groups. This was, for example, the case of array type X, shared by groups C and K, array type IX, shared by groups T and U, or array type V, shared by groups H, I, J, F and W (Figure 2A). Nevertheless, slight changes in the conformation of the knottin fold might have occurred in recent times, as suggested by the case of group N, whose members either displayed a type I or II array (Figure 1B, Appendix A). Phylogenetic inference also pointed out the close relationship between group E, which lost all cysteine residues except the Cys_3_-Cys_4_ pair, and other groups of canonical sCRP-I sequences (i.e., D, P and L).

#### 3.1.2. Molecular Diversity and Evolution of the mCRP-I Subfamily

The 16 mCRP-I groups, defined using the strategy reported in the materials and methods section, displayed highly conserved signal peptide and propeptide regions, followed by a mature peptide region characterized by the presence of a variable number of cysteine-rich arrays, ranging from two (groups ζ, η, θ, ξ and ο) to five (ν) (Figure 3). Unlike sCRP-I precursors, some mCRP-I groups (η, θ, κ, ξ and ο) also displayed a relatively long (i.e., 15–20 aa) C-terminal extension. The remarkable conservation of the signal peptide and propeptide regions was also observed in the degenerated group λ (not shown in the figure). Despite having lost all cysteine residues, group λ is likely evolutionarily linked with mCRP-I genes, as discussed in detail in Section 3.3. The 16 mCRP-I groups included a largely variable number of sequences, ranging from a single one (group ι), to 11 (group β). The full MSAs of the different groups are reported in Appendix A. Consistently with the variable number of associated cysteine arrays, mCRP-I mature peptides displayed widely different molecular weights, ranging from 6.37 KDa in ζ1, to 14.47 KDa in ν1. Nevertheless, they mostly retained their slightly cationic properties, with pI ranging from 7.2 in π1 to 8.8 in θ3 (data not shown).

Phylogenetic inference, supported with high confidence the identification of the 15 mCRP-I groups as monophyletic (Figure 4A; note that group λ was excluded from the MSA due to the lack of cysteine residues). The lone exception was the single sequence belonging to group ι, which, as discussed in detail below, was nearly identical to the members of the δ group, except for the lack of one cysteine array. The analysis evidenced a close relationship among groups α, β, γ, δ, ι, μ, ν and π. Similarly, the monophyly of the four groups characterized by the presence of just two cysteine arrays (i.e., η, θ, ξ and ο) was highly supported, suggesting a recent shared origin. Consistently with their unique cysteine array organization (Figure 3 and Figure 4), groups ε, ζ and κ were placed in three different clades, characterized by long branches, with high posterior probability support.

In spite of their size variation, mCRP-I peptides displayed just four cysteine array types (arrays II, VII, VIII and IX). Curiously, the N-terminal array was type VIII in all groups, except ζ and κ. (Figure 4A). This array also displayed the largest primary sequence diversity among groups, allowing their discrimination (Figure 4B). The following arrays were type VII, with the single exception of the second cysteine array of group ε (Figure 4A). Interestingly, several arrays found in different mCRP-I groups displayed a very high pairwise similarity, pointing out a shared evolutionary origin (Figure 4B). Based on the evidence gathered from the MSA, the presence of a different number of cysteine arrays in otherwise similar sequence groups might be the consequence of recent loss or acquisition events, as in the case of α and ν (see Figure 3). Curiously, based on the most parsimonious evolutionary interpretation, this acquisition/loss event of an additional cysteine array in this case did not involve a full knottin module, but happened in-between the third and fourth cysteine-rich array of ν. Similarly, δ and ι only differed due to the presence of either two or three cysteine arrays, with the inferred indel also occurring in-between the second and third cysteine module of δ (see Appendix A). These observations clearly suggest that an ongoing dynamic evolutionary process involving the tandem duplication of the knottin module might underpin the remarkable diversification of mCRP-I precursors. However, we have previously reported that sCRP-I and mCRP-I genes share the very same gene architecture, with the cysteine-rich mature peptide being invariably coded by the fourth and final exon, regardless of the number of cysteine arrays present [24]. Consequently, exon duplication and exon shuffling mechanisms, which are known to be the main drivers of diversification in other protein families with similar modular architectures [55,56], are not a plausible mechanistic explanation for the presence of a variable number of knottin modules in mCRP-I genes. We argue that the role of alternative processes, such as replication slippage, should be investigated in this context.

### 3.2. Each Mussel Individual Bears a Unique Repertoire of CRP-I Genes Due to Presence/Absence Variation

The analysis of available whole genome resequencing data from multiple individuals of three species belonging to the *Mytilus* species complex highlighted the presence of a high number of different complete (and thereby potentially functional) sCRP-I and mCRP-I genes in each individual. Moreover, multiple additional sequences characterized either by missing exons, frameshifts, nonsense mutations or by other mutations predicted to disrupt the knottin array were also detected. This observation was consistent with our previous report of the presence of CRP-I pseudogenes in the Mediterranean mussel [24], as well as with the general trend of recently expanded mussel gene families being associated with degenerated pseudogenes [13,28].

In total, each mussel genome displayed a number of unique sCRP-I sequences (defined as those encoding different mature peptides) more than double mCRP-I. In detail, ~20 sCRP-I (mean = 20.17, median = 20) and less than 10 mCRP-I sequences (mean = 7.13, median = 8) were identified in each genome (Figure 5). These figures are likely to be a slight underestimate of the real count, since some genes may not have been correctly assembled in some individuals due to the variable quality of available resequencing data. Nevertheless, we do not expect such differences to be particularly significant, as the number of sequences found in chromosome-scale genome assemblies were not much higher than mean and median counts.

In the absence of a chromosome-scale assembly for all the analyzed genomes, the different sequences belonging to the same sCRP-I and mCRP-I groups could not be unambiguously identified as allelic variants of the same gene or as recently duplicated paralogous genes. Even though phylogenetic inference could not solve these uncertainties (see Section 3.1.1), the frequent identification of three or more variants belonging to the same group in the same individual clearly indicated that a non-negligible fraction of these sequences were paralogous gene copies. This was further supported by the analysis of the CRP-I gene clusters in the fully-phased genome assembly of the blue mussel, as discussed in Section 3.3 below.

We chose to use a conservative approach, investigating gene PAV patterns in the 23 sCRP-I and 16 mCRP-I sequence groups outlined in Section 3.1.1 and Section 3.1.2 (Figure 5). A complete summary of PAV patterns based on the full set of the 113 sCRP-I and 77 mCRP-I sequences is provided in Appendix A. Overall, in line with the previously reported enrichment of genes encoding short, secreted, lineage-specific proteins in the dispensable fraction of the Mediterranean mussel genome, the CRP-I family was subjected to widespread gene PAV. Only three sCRP-I groups (i.e., G, H and V) were identified in all resequenced individuals and could therefore be considered as part of the core genome. All the others displayed a highly variable DI, which reached values as high as 0.96 in a few sequence groups only identified in a single individual (i.e., R, U and ι). This suggests that other divergent CRP-I sequences might be detected in the future by expanding the sample size and including other inter-fertile *Mytilus* species (e.g., *M. trossulus* or other mussels from the Southern hemisphere) in similar analyses. Interestingly, the mCRP-I subfamily was characterized, on average, by a much higher DI than sCRP-I, i.e., 0.57 vs. 0.37 (Figure 5). In line with observations collected for other mussel immune effectors, on several occasions, the lack of functional genes was paired with the observation of a pseudogene belonging to the same sequence group (Figure 5) [28].

Although no marked difference in PAV profiles and CRP-I sequence number was immediately evident among *M. galloprovincialis*, *M. edulis* and *M. chilensis*, the low number of individuals available for the two latter species currently prevents drawing definitive conclusions about the association between PAV patterns and species phylogeny for this gene family.

### 3.3. Organization of the CRP-I Gene Clusters in M. edulis

The accurate annotation of CRP-I genes in the blue mussel reference genome allowed the identification of four distinct gene clusters. Three of these were placed in chromosome 5, separated by approximately 2.3 and 13.5 Mb of sequence, respectively, whereas the fourth one was found in chromosome 7 (Figure 6). In total, we identified 17 complete sCRP-I genes and 12 mCRP-I genes. However, the subsequent analysis of the two separate haplotype assemblies allowed the identification of five additional CRP-I genes, bringing their total number to 22. The blue mussel reference genome did also include a considerable number (i.e., 14 sCRP-I and 11 mCRP-I) of pseudogenes, which were generally mixed with functional genes, suggesting the frequent accumulation of deleterious mutations in paralogous gene copies.

Cluster one, the largest out of the four, spanned ~540 Kb and included 6 complete sCRP-I and 11 complete mCRP-I genes, respectively, plus a total of 18 pseudogenes (11 belonging to the sCRP-I subfamily, 7 belonging to the mCRP-I subfamily). The phasing of the *M. edulis* genome did not allow us to obtain the complete sequence of the two cluster one haplotypic variants, as the reference genomic sequence appeared to be a combination of the two (Figure 6). Cluster two spanned ~300 Kb in the reference assembly and included six sCRP-I and six mCRP-I genes, respectively, plus a total of six pseudogenes (two belonging to the sCRP-I subfamily, four belonging to the mCRP-I subfamily). As in the case of cluster one, chromosome phasing was not successful in retrieving the two complete haplotypic variants. However, the analysis of the scaffolds allowed extending this gene cluster at both ends. In detail, an additional sCRP-I pseudogene (belonging to group V) was found neighboring the B1 gene at one extremity of the cluster. Another degenerated pseudogene and two full sCRP-I genes (i.e., O2 and H1) were detected at the other end of the cluster, extending it by over 50 Kb (Figure 6). The organization of the other two CRP-I gene clusters was much simpler. Cluster three only included two sCRP-I genes, i.e., H2 and H8, found in chromosome 5 in a 15 Kb-long region. In haplotype 2, H2 was replaced by H1, a highly similar variant (Figure 6). Cluster four, the only one to be placed on chromosome 7, comprised of three sCRP-I genes (C3, K3 and K4), found in a 75 Kb-long region in the reference genome assembly. However, the architecture of this cluster was simpler in the separated haplotype assemblies, which only included two genes (K3 was shared by both, Figure 6). Finally, three additional sCRP-I genes that were not present in the reference assembly (i.e., A1, G3 and O2) were detected in unplaced genomic scaffolds in haplotype 2 and may therefore correspond to non-scaffolded contigs found in a hemizygous state in the individual used to obtain the genome assembly.

Fourteen sCRP-I groups (i.e., A, B, C, F, G, H, I, J, K, M, N, O, P and V) were represented by at least one gene in the blue mussel genome and an additional one (i.e., Q) was only found as a non-functional pseudogene. The other eight sequence groups (i.e., D, E, L, R, S, T, U and W) were not detected. These results are fully consistent with the gene PAV patterns observed, on a larger scale, in the *Mytilus* species complex (see Figure 5), as all the missing sequence groups were characterized by a DI higher than 0.4. Similar results were obtained for the mCRP-I subfamily, as 10 out of the 16 sequence groups were represented by complete genes (i.e., groups α, β, γ, ε, ζ, η, θ, λ, μ and ξ). δ, κ and ν were only found as pseudogenes, whereas no ι, ο or π genes were detected. As in the case of sCRP-I genes, most missing sequence groups (except ο) had a DI higher than 0.4 (Figure 5). The genomic analysis also confirmed that the sequence groups we defined did not include only putative allelic variants, but also closely-related paralogs. For example, two identical η1 genes were found tandemly duplicated in cluster one, and two identical I1 genes were present in clusters one and two. On other occasions, slightly different sequences belonging to the same group were found in close proximity to each other, also suggesting a recent origin by tandem duplication: this was the case, for example, for genes V1 and V2 in cluster one, H1 and H2 in cluster three, or K3 and K4 in cluster four. As of note, a few highly divergent pseudogenes (indicated with “?” in Figure 6) could not be unambiguously assigned to any sequence group. The genes found in cluster four, the only one located on chromosome 7, belonged to groups C and K, were the only ones to show the peculiar type X cysteine array, and shared other unique features, such as the presence of an Arg–Arg dibasic proprotein convertase cleavage site (Figure 1A). Altogether, these observations suggest that groups C and K may have diverged very early from all other CRP-I genes, upon translocation to a different chromosome.

Despite being significantly different in terms of number of genes and size, clusters one and two displayed a similar general organization, with mCRP-I genes found in the central part, flanked at both sides by sCRP-I genes. This, together with the small distance that often separated sCRP-I genes with their neighboring mCRP-I genes, strongly supports the close evolutionary relationships between these two subfamilies. Moreover, the placement of λ genes among mCRP-I genes and pseudogenes in both cluster one and in cluster two supports their classification within this CRP-I subfamily, notwithstanding the lack of cysteine residues, which is interpretable as a secondary loss.

As we have previously demonstrated, *Mytilus* is characterized by a highly unusual pan-genomic architecture, with widespread occurrence of hemizygosity [13]. Since the standard protocols for de novo genome assembly, designed and mostly applied to species with relatively low heterozygosity rates, lacking massive structural variation, aim at producing a monoploid reference assembly, a significant fraction of inter-haplotype diversity might have been lost in the PEIMed reference. As reported above, the investigation of the architecture of the four CRP-I loci in the phased assemblies of both haplotypes was not always successful in recovering the complete sequences of the two homologous chromosomes. The high complexity of *Mytilus* genomes might therefore still represent a significant obstacle towards the correct assembly of large genomic regions characterized by extreme PAV, despite the availability of high-quality long reads and chromosome conformation capture libraries.

### 3.4. The sCRP-I and mCRP-I Subfamilies Display Different Expression Levels and Tissue Specificities

We have previously investigated the trends of expression of a few target sCRP-I and mCRP-I sequences using qRT-PCR, evidencing highly variable profiles, which nevertheless suggested that these genes were expressed at higher levels in the mantle and digestive gland tissue, compared with hemocytes, gills, posterior adductor muscle and foot [24]. However, in light of the evidence reported in Section 3.2, these preliminary results were likely affected by primer design and widespread gene PAV, which altogether might have impaired target amplification in some individuals. Aiming at providing a comprehensive overview of the tissue specificity of all CRP-I genes, we exploited the availability of 215 RNA-seq datasets to assess the expression levels of all sCRP-I and mCRP-I genes in the Mediterranean mussel. The analyzed transcriptomic data were characterized by largely diverse geographical locations of sampling, genomic backgrounds, environmental and experimental conditions, and were therefore expected to display significant heterogeneity. To take this into account, gene expression levels will be here reported as the mean and maximum level of expression observed across all samples in four key tissues, i.e., mantle, gills, digestive gland and hemocytes, and in larvae at the mid-trochophora stage.

The cumulative transcription of all the members of the CRP-I superfamily often reached significant levels. In detail, the sum of all sCRP-I genes reached a mean expression value of ~1220 TPM in mantle, ~310 TPM in the digestive gland, ~35 TPM in hemocytes and ~20 TPM in gills (Figure 7A). The expression of mCRP-I genes was significantly lower in mantle (i.e., ~55 TPM), digestive gland (~20 TPM) and gills (~6 TPM), but higher in hemocytes (~100 TPM) (Figure 7B). The maximum observed expression levels were in line with these trends. Indeed, the max observed cumulative expression level of all sCRP-I genes was close to 10,000 TPM (i.e., 1% of total mRNA transcription) in mantle, 4200 TPM in the digestive gland, ~230 TPM in hemocytes and ~180 TPM in gills. The highest expression levels observed for the mCRP-I subfamily were ~630 TPM in hemocytes, ~400 TPM in mantle, ~100 TPM in digestive gland and ~90 TPM in gills. Hence, while being consistent with our previous qRT-PCR data in identifying the mantle and digestive gland as the primary sites of sCRP-I transcription, these data clearly highlighted the presence of significant differences with mCRP-I genes, which were cumulatively expressed at higher levels in hemocytes and mid-trochophora larvae (Figure 7B).

A finer-scale analysis allowed an in-depth investigation of the expression of individual sequence groups, which were highly heterogeneous. Such differences were explained only in part by gene PAV, even though several groups with high DI (see Figure 5) were expressed in a very low fraction of samples, with the extreme cases of groups R, W and ξ being undetected in all the 215 analyzed transcriptomic datasets. Other sequence groups, such as the sCRP-I groups A, J and H, were expressed in 82%, 71% and 57% of all samples, respectively. Within the mCRP-I subfamily, groups π and η were expressed in 67% and 58% of samples, respectively (Figure 7C). In terms of mean expression levels, five sCRP-I sequence groups reached 100 TPM (i.e., 0.01% of the total transcriptional effort of a given tissue) in the mantle (C, D, G, J and K), two in the digestive gland (H and J) and none in the gills, hemocytes and larvae (Figure 7D). The same sequence groups exceeded 1000 TPM (i.e., 0.1% of the total transcription) in a few individual samples, with the maximum levels being recorded for groups C and J (>6000 TPM) in mantle (Figure 7E). Among mCRP-I genes, only π achieved a mean expression level higher than 100 TPM in larvae, but η and κ exceeded this value in a few hemocyte samples. Globally, the overwhelming majority of both sCRP-I and mCRP-I sequence groups displayed a marked specificity of expression in either the mantle, the digestive gland, or both tissues. Nevertheless, a few groups were preferentially expressed in the hemocytes (e.g., κ) or in larvae (e.g., π) (Figure 7F).

In summary, while the cumulative expression of all CRP-I genes was not negligible in the mantle and digestive gland, several sequence groups displayed very low and largely variable expression levels. Altogether, these observations suggest that CRP-I genes may be expressed by specialized cell types with low relative abundance in the macrotissues that are usually sampled for RNA-sequencing approaches. An alternative explanation could reside in the specific expression of CRP-I genes in minor tissues that could not be analyzed in this study due to the lack of available RNA-seq datasets, even though we have previously demonstrated the lack of significant expression in the foot and posterior adductor muscle [24]. An additional factor that should be taken into account is linked with the regulatory routes underpinning CRP-I gene expression, which are yet to be unveiled. Indeed, the expression of these genes may be triggered in response to unknown stimuli.

### 3.5. CRP-I Peptides Belong to the Knottin Fold Superfamily

The remarkable advancements in ab initio protein three-dimensional structure prediction though the implementation of artificial intelligence now allows obtaining highly accurate structural models even for protein sequences lacking suitable templates for homology modeling [57]. By using AlphaFold v2.3.0 [49,50], we here provide an overview of the predicted three-dimensional structure of CRP-I peptides. Although we had previously speculated that these peptides were likely to adopt a knottin-like fold, three peptides obtained by solid-phase synthesis in our previous study displayed a disulfide bond topology (i.e., Cys_1_-Cys_2_, Cys_3_-Cys_4_, Cys_5_-Cys_6_) inconsistent with such predictions [24]. Nevertheless, we speculate that this discrepancy was due to the lack of disulfide isomerase activity, which accelerate the kinetics of the formation of the complex patterns of disulfide bonds found in numerous animal cysteine-rich peptides [58].

In general, the predictions obtained with AlphaFold were highly concordant in identifying the knottin fold as the best supported structural architecture for the vast majority of sCRP-I peptides, with per-target pLDDT generally achieving good confidence scores, comprised between 70 and 90 (Appendix A). Despite the high primary sequence diversity of the mature peptides (see Section 3.1.1), and notwithstanding the significant differences among disulfide arrays (Figure 1B), all sCRP-I sequences converged, for at least one member of each group, on the knottin fold as the best supported one. The predicted structures evidenced a highly conserved backbone, with minor differences in the N- and C-terminal ends (Figure 8A), consistently with the lower confidence scores generally obtained for these regions (Appendix A). The majority of the few sCRP-I sequences that did not agree with such predictions (at least for the top-scoring model) belonged to group B, the only one to display an unusually short loop b. The presence of the knottin fold strongly supported the Cys_1_-Cys_4_, Cys_2_-Cys_5_, Cys_3_-Cys_6_ disulfide connectivity (Figure 8B), with all predicted bonds showing lengths comprised between 1.97 Å and 2.05 Å. The multiple cysteine-rich modules found in mCRP-I mature peptides were also predicted, in all cases, to be arranged in a knottin fold, with high confidence scores (Appendix A). Albeit further proteolytic processing of mCRP-I peptides into smaller peptides comprising a single knottin module cannot be excluded, the precursors were predicted to adopt, upon pro-peptide cleavage, a pearl necklace-like architecture (Figure 8C).

In line with such observations, the structural comparisons performed with Dali [53] identified a close resemblance to several previously characterized knottins. These included several toxins from spiders (e.g., purotoxin-1 and -6 [59], psalmotoxin-1 [60], J-atracotoxin-HV1C [61] and ceratoxin-1 [62]), cone snails (e.g., the conotoxins GXIA [63], GS [64]) and scorpions (e.g., the U1-liotoxin-Lw1a [65]). Nevertheless, other close structural matches were AMPs [66] or protease inhibitors [67,68], delineating the broad possible functional range of CRP-I mature peptides.

### 3.6. The sCRP-I H1 Peptide Does Not Show Significant Antimicrobial or Protease Inhibitor Activities, but May Act as a Defense Peptide

The sCRP-I H1 peptide was selected as a representative member of the sCRP-I subfamily for solid phase synthesis and functional characterization. H1 belongs to one of the two core sCRP-I groups (Figure 5), displays a DI = 0.52 and was highly expressed in both the digestive gland and mantle tissues (Figure 7). The peptide was synthesized respecting the predicted knottin disulfide connectivity, thereby avoiding the formation of anomalous disulfide bonds that occurred in our previous solid synthesis approach [24]. Moreover, the peptide was C-terminally amidated (APCWPRGCFRDRDCCYGYQCSYRKCMRKR-NH_2_), reflecting one of the most frequently observed post-translational modifications occurring in short molluscan bioactive peptides [54]. As of note, this modification could be only predicted in-silico and should be validated through the isolation of the peptide from tissue extracts, which may be a challenging task due to gene PAV (see Section 3.2) and the high variability of expression of CRP-I genes (see Section 3.4).

We evaluated the possible role played by this peptide, and by extension the other members of the CRP-I superfamily, based on the evaluation of the most commonly reported biological activities of knottins in the literature. As some knottins act as broad-spectrum protease inhibitors [69], we tested whether the H1 peptide could inhibit the activity of four proteases representative of different classes, i.e., papain (cysteine protease), thermolysin (metalloprotease), pepsin (aspartate protease) e subtilisin A (serine protease). Nevertheless, at the tested peptide concentration (1.37 × 10^−6^ M) no significant inhibition of activity was evident for any of these representatives of the four classes of proteases (Table 1), discouraging, in the frame of a preliminary characterization, the hypothesis that the H1 peptide was endowed with this biological activity.

We subsequently tested the antimicrobial activity of H1, with a MIC assay, against four bacterial species, i.e., *E. coli*, *P. aeruginosa*, *E. faecalis* and *S. aureus*, at concentrations up to 32 µM, since several AMPs belonging to the knottin superfamily have been previously reported in other organisms, including invertebrate metazoans [70,71]. After 24 h of incubation with the peptide, the growth of none of the four tested bacterial species was affected under standard laboratory conditions (Table 1), discouraging further investigations in this direction and suggesting that CRP-I peptides lacked significant antimicrobial activity under the tested conditions. Nevertheless, we presently cannot exclude the possibility that these peptides retain significant activity under marine-like conditions, or towards non-cultivable bacteria untested in the present experiment, but abundant in the marine environment.

The vast majority of the knottins previously identified in invertebrates are toxins, usually recruited as components of venom for predation. Among these, conotoxins [72], spider [73] and scorpion neurotoxins [74] are certainly among the most well-known cases, even though the venoms of other invertebrates have acquired the same structural fold in a convergent manner [75]. We preliminarily tested this toxicity of sCRP-I H1 by: (i) running an in vitro MTT assay on SH-SY5Y cell lines to assess the effects of the peptide on eukaryotic cell metabolic activity after 24 h of exposure and (ii) carrying out an in vivo toxicity test by injecting *G. melonella* larvae with different peptide concentrations and evaluating larval mortality at 24 and 48 h post injection.

The MTT assay supported the possibility that the sCRP-I H1 peptide was cytotoxic towards eukaryotic cells, as revealed by the significant reduction in metabolic activity observed after 24 h of exposure at concentrations equal to or higher than 10 μM (Table 1). The results of the in vivo assay suggested that the H1 peptide may exert, to some extent, a toxic activity in the *G. melonella* system, due to the increased mortality rate observed in particular 48 h post injection, which reached 45% and 63% in the groups injected with 30 and 300 μg/Kg peptide, respectively (Table 1).

The two tested biological systems were largely divergent from the most plausible eukaryotic targets of CRP-I peptides. Nevertheless, all eukaryotes are expected to share highly conserved membrane-bound channels that could represent the biological targets of knottins in the venoms of other animals. Under the assumption that the primary function of CRP-I peptides is immune defense, such target channels may be associated with eukaryotic parasites, which have been frequently reported in literature as responsible for bivalve pathologies. These include, among others, Ascetospora and Apicomplexa protozoans, which often cause massive mortalities in different bivalve species [76,77,78], trematode and cestode flatworms [79], or even parasitic crustaceans [80].

Altogether, even though some preliminary indications suggest that CRP-I peptides may act as toxins towards unidentified eukaryotic parasites, we cannot rule out the possibility that they carry out different biological functions that we could not explore within the frame of the present work. Moreover, knottins are frequently subjected to rare post-translational modifications, undetectable by bioinformatics means, that could not be included in the tested H1 synthetic peptide but may have a significant impact on their function [81].

## 4. Conclusions

The complex organization of the four CRP-I loci found in the blue mussel reference genome, together with the data obtained though the comparative genomic analysis of 23 individuals belonging to different species of the *Mytilus* complex, support the occurrence of widespread gene PAV within this family. We provide a significant update to the repertoire of mussel CRP-I peptides, identifying over 100 different unique mature sequences, that could be classified within 23 sCRP-I and 16 mCRP-I subfamilies, respectively. While the complex evolutionary history of this gene family might have led to a remarkable functional diversification in parallel with the acquisition of novel gene copies due to tandem duplications and subsequent positive selection, the role of CRP-I peptides remains elusive. Through the functional characterization of the sCRP-I H1 peptide, we propose that this gene family may be implicated in innate immune defense, acting against invading eukaryotic parasites at the main sites of production, i.e., in the mantle and gill tissues.

## Figures and Tables

**Figure 1 genes-14-00787-f001:**
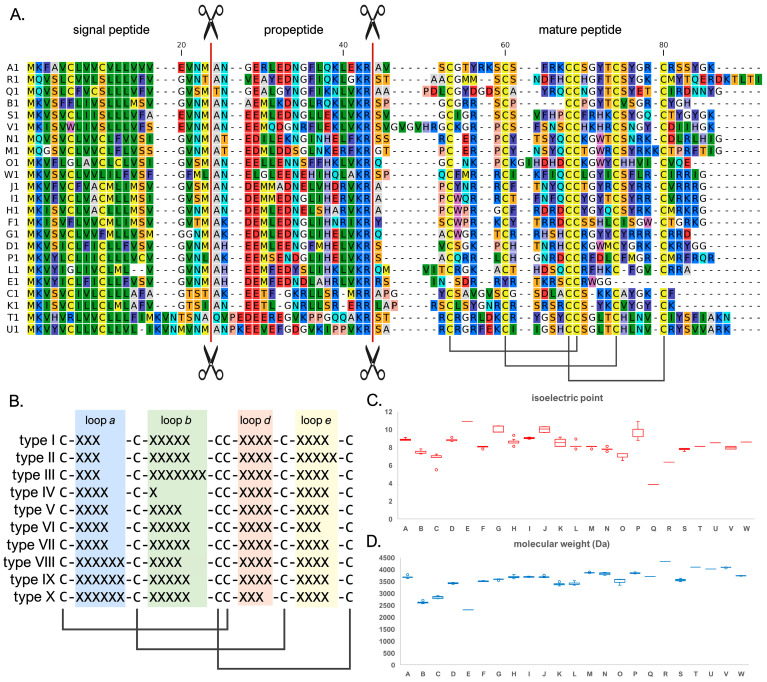
Panel (**A**): multiple sequence alignment of the full-length protein precursors of representative members of each of the 23 sCRP-I sequences. The inferred position of the signal peptide and propeptide cleavage sites are marked by scissors. The inferred disulfide connectivity between the six highly conserved cysteine residues involved in the establishment of the knottin fold is indicated at the bottom of the mature peptide region. Panel (**B**): schematic representation of the ten types of cysteine arrays characterizing sCRP-I peptides. Loop a, b, d and e indicate the four loops connecting conserved cysteine residues in the primary sequence of knottins. Note that loop c is missing due to the neighboring position of Cys3 and Cys4. Panel (**C**): isoelectric point (mean plus standard deviation, outliers are indicated by circles) of the mature peptide sequences of each sCRP-I group (indicated by uppercase letters). Panel (**D**): molecular weight (mean plus standard deviation, outliers are indicated by circles) of the mature peptide sequences of each sCRP-I group (indicated by uppercase letters).

**Figure 2 genes-14-00787-f002:**
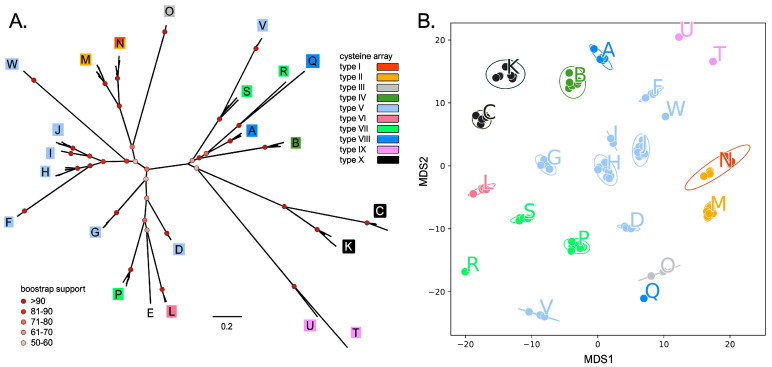
Panel (**A**): maximum likelihood unrooted phylogenetic tree of mussel sCRP-I sequences. Each of the 23 groups is indicated with a letter, and the background color refers to one of the ten types of cysteine arrays (see Figure 1B). For simplicity’s sake, only the bootstrap support values of the major nodes of the tree are displayed with colored circles. Panel (**B**): MDS plot obtained from the pairwise Hamming distances between the sCRP-I sequences reported in this study. Each dot marks a sequence and each group of sequences is colored based on the type of characterizing disulfide array (see Figure 1B).

**Figure 3 genes-14-00787-f003:**
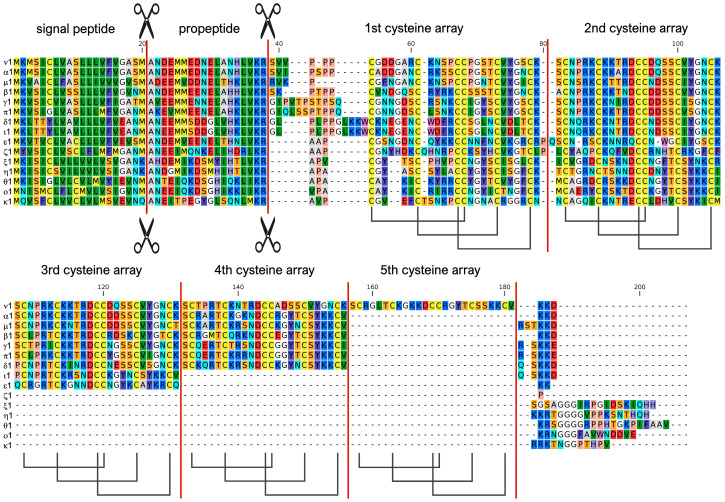
Multiple sequence alignment of the full-length protein precursors of representative members of each of the 15 mCRP-I sequence groups reported in this study (excluding group λ, which entirely lacks conserved cysteine residues). The inferred position of the signal peptide and propeptide cleavage sites are marked by scissors and by a vertical bar, which also indicates the boundaries among the different knottin arrays. The inferred disulfide connectivity between the six highly conserved cysteine residues involved in the establishment of the knottin fold is indicated at the bottom of the mature peptide region.

**Figure 4 genes-14-00787-f004:**
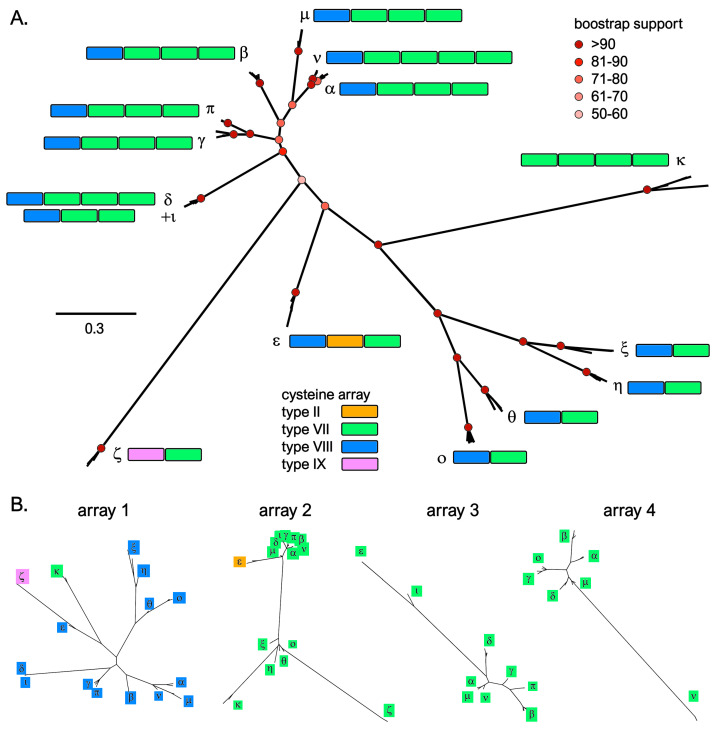
Panel (**A**): maximum likelihood unrooted phylogenetic tree of mussel mCRP-I sequences. Each of the 15 groups of sequences are indicated with a Greek letter, and the number and organization of cysteine arrays are indicated by the presence of color. For simplicity’s sake, only the bootstrap support values of the major nodes of the tree (i.e., those defining the 15 sequence groups, or more basal ones) are displayed. Panel (**B**): maximum likelihood unrooted phylogenetic trees based on the multiple sequence alignment of the four cysteine arrays found in mCRP-I peptides; array 5, exclusively present in group ν, was not included.

**Figure 5 genes-14-00787-f005:**
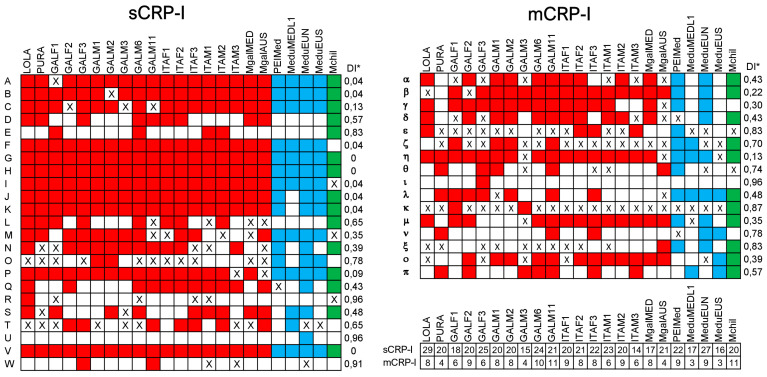
Patterns of gene presence/absence variation for complete and potentially functional sCRP-I and mCRP-I genes (i.e., excluding pseudogenes) in different sequenced individuals of *M. galloprovincialis* (red), *M. edulis* (blue) and *M. chilensis* (green). See Section 2 for a detailed description of the analyzed individuals. Colored and white boxes indicate the presence, or the absence of a given sequence group in each individual. “X” marks the identification of a pseudogene (but no functional gene) belonging to a given sequence group. A table summarizes the total number of unique complete sCRP-I and mCRP-I genes identified in each individual. * DI: dispensability index, which indicates the fraction of analyzed individuals where any given sequence group was absent.

**Figure 6 genes-14-00787-f006:**
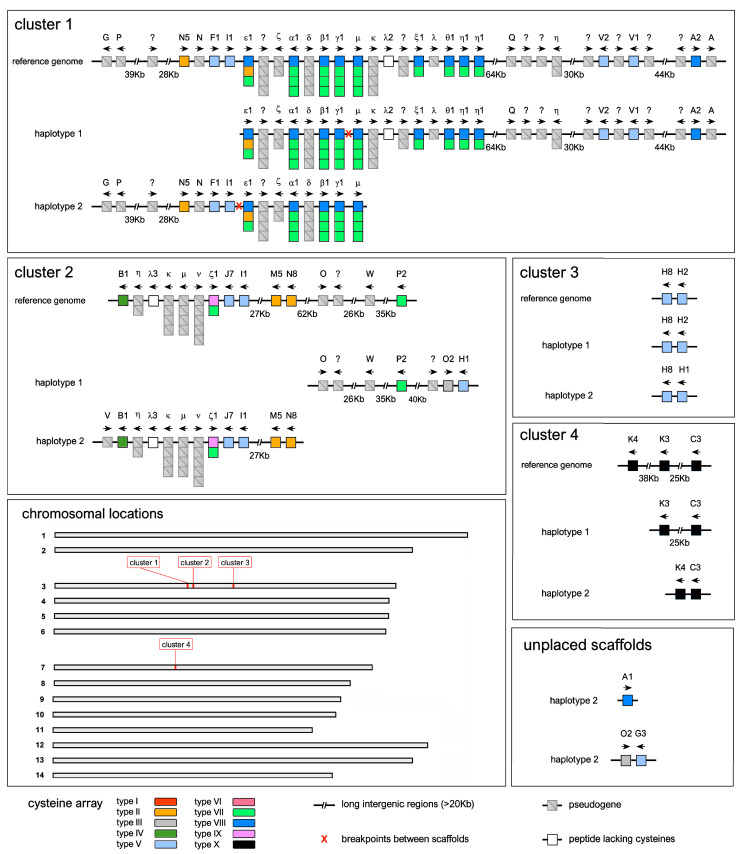
Organization and chromosomal location of the four CRP-I gene loci, with a detailed overview of the gene and pseudogene annotations present in the reference genome assembly, as well as in the two distinct phased haplotype assemblies. Each sCRP-I gene is indicated by a single box, whereas mCRP-I genes are indicated by multiple stacked boxes, representing, from the top to the bottom, N-terminal and C-terminal knottin arrays. Pseudogenes are marked by crossed-out grey boxes. Arrows indicate the coding strand of each gene (forward or reverse).

**Figure 7 genes-14-00787-f007:**
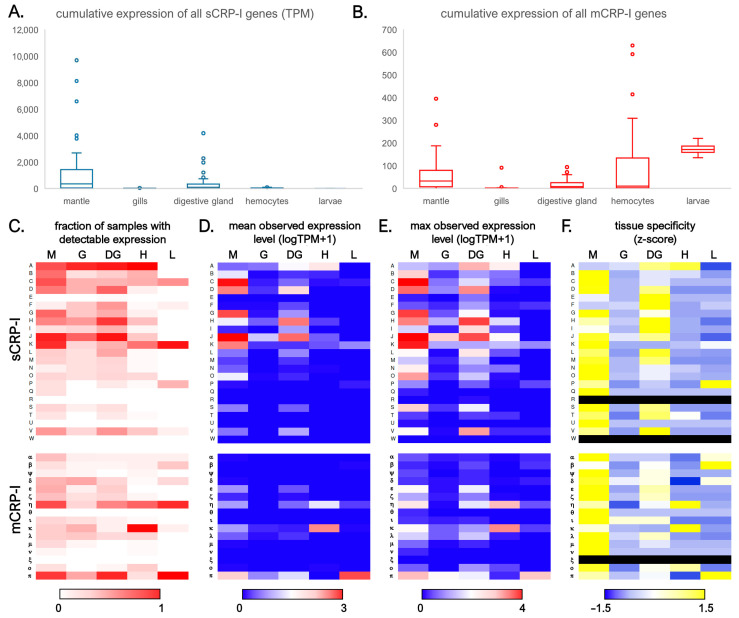
Panel (**A**): range of cumulative expression levels of sCRP-I genes in publicly available transcriptomic datasets of *M. galloprovincialis*; outlier samples are indicated by circles. Panel (**B**): range of cumulative expression levels of mCRP-I genes in publicly available transcriptomic datasets of *M. galloprovincialis*. Panel (**C**): fraction of samples from each of tissue where the expression of any sequence groups was detected (i.e., TPM > 0). Panel (**D**): mean TPM expression level observed across all samples from each tissue. Panel (**E**): maximum TPM expression level observed across all samples from each tissue. Panel (**F**): tissue specificity, displayed as the z-score for all sCRP-I and mCRP-I sequence groups; black lines indicate sequence groups lacking detectable expression in all samples. M: mantle; G: gills; DG: digestive gland; H: hemocytes; L: larvae at the mid-trochophora stage.

**Figure 8 genes-14-00787-f008:**
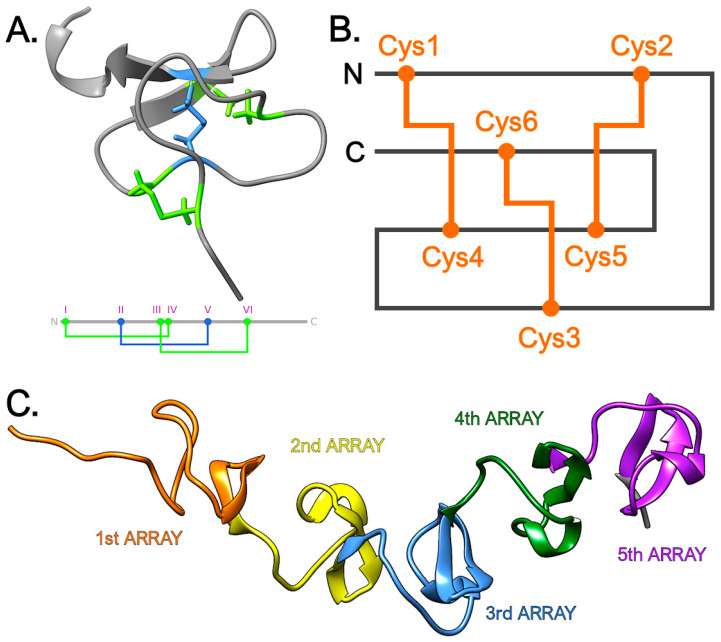
Panel (**A**): predicted 3D structure of sCRP-I A3, with a simplified view of the inferred disulfide connectivity, plotted on the linearized sequence of the peptide. Panel (**B**): exemplified typical disulfide array of the knottin superfamily, predicted to be present also in the CRP-I family. Panel (**C**): “pearl necklace” structural organization of a mCRP-I mature peptide, here exemplified by mCRP-I ν1.

**Table 1 genes-14-00787-t001:** Summary of the outcomes of the functional assays carried out on the sCRP-I H1 peptide. NS: not statistically significant.

protease Inhibition		
protease tested	protease class	inhibitory activity
papain	cysteine protease	NS
thermolysin	metalloprotease	NS
pepsin	aspartate protease	NS
subtilisin A	serine protease	NS
antimicrobial activity		
bacterial strain	incubation time	MIC
*Escherichia coli* ATCC 25922	24 h	>32 µM
*Pseudomonas aeruginosa* ATCC 27853	24 h	>32 µM
*Enterococcus faecalis* ATCC 19433	24 h	>32 µM
*Staphylococcus aureus* ATCC 25923	24 h	>32 µM
in vivo cytotoxicity in *Galleria melonella*		
peptide concentration	incubation time	mortality
untreated control (PBS)	24 h	1%
30 μg/Kg	24 h	4%
300 μg/Kg	24 h	12%
untreated control (PBS)	48 h	13%
30 μg/Kg	48 h	45%
300 μg/Kg	48 h	63%
in vitro cytotoxicity in the SH-SY5Y cell line		
peptide concentration	incubation time	cell viability *
SH-SY5Y (0.5 μM)	24 h	−1.93%
SH-SY5Y (1 μM)	24 h	−2.67%
SH-SY5Y (10 μM)	24 h	−34.81%
SH-SY5Y (25 μM)	24 h	−41.42%
SH-SY5Y (50 μM)	24 h	−31.60%
SH-SY5Y (100 μM)	24 h	−42.10%

* Drop of cell viability with respect to an untreated control.

## Data Availability

All genomic and transcriptomic data analyzed in this study are publicly available in the NCBI Genomes and NCBI SRA databases.

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
