# Peer review of "Unveiling the Impact of Gene Presence/Absence Variation in Driving Inter-Individual Sequence Diversity within the CRP-I Gene Family in Mytilus spp."

_genes, 2023, doi:10.3390/genes14040787_

Round 1
Reviewer 1 Report
This manuscript descibed the short secretory cationic cysteine-rich peptides (CRP-I) in Mytilus spp. Overal merit of this manuscript is good and is deserved publication. There are only some minor spell errors existed needing revision.
Author Response
This manuscript descibed the short secretory cationic cysteine-rich peptides (CRP-I) in Mytilus spp. Overal merit of this manuscript is good and is deserved publication. There are only some minor spell errors existed needing revision.
We thank the reviewer for his/her positive assessment of our manuscript. The text has been carefully reviewed to fix minor spelling errors.
Reviewer 2 Report
Comments genes-2254793
Manuscript genes-2254793 is a work profiling molecular characterization of CRP-1 gene and its biological functions in Common mussel (Mytilus edulis). Ms is well written and molecular data is comprehensively presented. However, some issues with minor category are found in the text.
General comments
- Modify the title! It is not common to use a complete sentence for a title.
- Use the common name to mention species, and their Latin name is only included at the first appearance of the species. Apply this throughout the text!
Specific comments
Title
4-5 include the common name for Mytilus spp!
Abstract
22 – a) characterized, and b) state the common name for Mytilus edulis
24 – reported
Introduction
100 – data have
Materials and methods
264-9 – state all the housekeeping genes used here!
Conclusions
867-74 – This information is not necessary to be included here, they are just like repetitions from the introduction. Focus only on the points resulting from the present work in relation to addressing the research objectives.
Author Response
Manuscript genes-2254793 is a work profiling molecular characterization of CRP-1 gene and its biological functions in Common mussel (Mytilus edulis). Ms is well written and molecular data is comprehensively presented. However, some issues with minor category are found in the text.
We thank the reviewer for his/her positive assessment of our manuscript.
General comments
- Modify the title! It is not common to use a complete sentence for a title.
Based on the suggestion provided by the referee, we have modified the title as follows: “Unveiling the impact of gene presence/absence variation in driving inter-individual sequence diversity within the CRP-I gene family in Mytilus spp.”
- Use the common name to mention species, and their Latin name is only included at the first appearance of the species. Apply this throughout the text!
We thank the reviewer for this suggestion. However, the common names used for different mussel species (both within the Mytilus genus and for other mussel genera) largely vary depending on local denominations and may be a source of confusion. The term “mussel”would be way too generic, since it is generally used to design also members of the order Mytiloida belonging to the Perna genus (e.g. the greenshell mussel P. viridis), to the Bathymodioulus genus (e.g. several species of deep sea hydrothermal vent mussels) and many others. Moreover, the same term is incorrectly also used to indicate the freshwater zebra and quagga mussels belonging to the Dreissena genus.This situation is made even more complex by the uncertain reproductive boundaries among Mytilus species, which de facto belong to a species complex. Altogether, these considerations lead most authors to preferentially use scientific names instead of common names.
Nevertheless, we recognize that the excessive use of scientific names may be reduced and we have consequently carefully revised the text, replacing scientific names with common names on a few occasions, whenever this use could not lead to ambiguities.
Specific comments
Title
4-5 include the common name for Mytilus spp!
See our previous comment above. Since this study is focused on multiple Mytilus species and considering the fact that the term “mussels” can be ambiguous, we prefer to use “Mytilus spp” in the title.
Abstract
22 – a) characterized, and b) state the common name for Mytilus edulis
Thank you, this has been corrected.
24 – reported
Thank you, this has been corrected.
Introduction
100 – data have
Thank you, this has been corrected.
Materials and methods
264-9 – state all the housekeeping genes used here!
Thank you, the definition of housekeeping genes was added, along with their accession ID.
Conclusions
867-74 – This information is not necessary to be included here, they are just like repetitions from the introduction. Focus only on the points resulting from the present work in relation to addressing the research objectives.
Thank you for this suggestion. Those sentences have been deleted.